# Graded intrafillable architecture-based iontronic pressure sensor with ultra-broad-range high sensitivity

Ningning Bai[1,4], Liu Wang [1,2,4], Qi Wang[1], Jue Deng[1,2], Yan Wang[1], Peng Lu[1], Jun Huang[1], Gang Li[1], Yuan Zhang[1], Junlong Yang[1], Kewei Xie[1], Xuanhe Zhao [2] & Chuan Fei Guo [1,3]*

Sensitivity is a crucial parameter for flexible pressure sensors and electronic skins. While introducing microstructures (e.g., micro-pyramids) can effectively improve the sensitivity, it in turn leads to a limited pressure-response range due to the poor structural compressibility. Here, we report a strategy of engineering intrafillable microstructures that can significantly boost the sensitivity while simultaneously broadening the pressure responding range. Such intrafillable microstructures feature undercuts and grooves that accommodate deformed surface microstructures, effectively enhancing the structural compressibility and the pressure-response range. The intrafillable iontronic sensor exhibits an unprecedentedly high sensitivity ($S_{min} > 220$ kPa$^{-1}$) over a broad pressure regime (0.08 Pa-360 kPa), and an ultrahigh pressure resolution (18 Pa or 0.0056%) over the full pressure range, together with remarkable mechanical stability. The intrafillable structure is a general design expected to be applied to other types of sensors to achieve a broader pressure-response range and a higher sensitivity.

---

[1] Department of Materials Science and Engineering and Centers for Mechanical Engineering Research and Education at MIT and SUSTech, Southern University of Science and Technology, 518055 Shenzhen, China. [2] Department of Mechanical Engineering, Massachusetts Institute of Technology, Cambridge, MA 02139, USA. [3] Shenzhen Engineering Research Center for Novel Electronic Information Materials and Devices, Southern University of Science and Technology, 518055 Shenzhen, Guangdong, China. [4] These authors contributed equally: Ningning Bai, Liu Wang. *email: guocf@sustc.edu.cn

Flexible pressure sensors and electronic skins (e-skins) have attracted much interest because of their capability to sense mechanical stimuli and have thus been envisioned as key technologies for the applications of health monitoring[1–3], artificial intelligence[4,5], and human-machine interfaces[6–8]. A capacitive pressure sensor (CPS), a device that consists of two electrodes sandwiching a soft dielectric between them, transduces pressure stimuli to capacitance signals. While CPSs often present advantages of high-drift stability and a simple structure, and are considered as a promising selection for high-performance flexible pressure sensing[9–13], they exhibited limited sensitivity ($S$) and low-pressure resolution over a broad pressure range or a saturated response at high pressures. A conventional piezo-CPS usually employs a blanket dielectric film that is incompressible and viscoelastic, resulting in the device exhibiting low-sensitivity together with low-response speed. Engineering the dielectric film with microstructured surfaces has proven to be an effective method to improve both sensitivity and response speed[14–16]. These applied microstructured surfaces include a wide variety of topographical structures such as micro-pyramid arrays[1,3,16,17], wrinkles[18,19], micro-domes[20], micro-pillar arrays[21,22], and other cone-like patterns directly molded from natural prototypes such as plant leaves[23–25]. Such structures, however, are mechanically stable upon compression, and thus respond effectively only under low pressures. As a result, this strategy fails to produce an adequately high sensitivity for sensors over a wide pressure range ($S_{max} < 2 \text{ kPa}^{-1}$)[16,21,26,27]. Recently, incorporating an elastomer with ionic liquid as the dielectric has emerged as a method for further enhancing the sensitivity[25,28–31]. By forming electron double layers (EDLs) at the dielectric/electrode interface[32,33], the capacitance of iontronic devices can be remarkably elevated due to the atomic scale distance (~1 nm) between positive and negative charges at the EDL interface, significantly promoting the piezo-capacitive effect upon compression. While both high sensitivity and high-pressure resolution at pressures over 100 kPa are demanded for various applications such as robotic manipulation, pressure monitoring in human body, and pressure tests in high-speed fluids, current microstructured CPSs, including supercapacitive iontronic e-skins and sensors, suffer from limited or saturated response under high pressures (>100 kPa)[20,25,29,30,34,35]. The unreconciled tradeoff between high sensitivity and a broad working range of pressures for flexible pressure sensors (not limited only to capacitive-type sensors) is attributed to the low compressibility of the stable microstructures, along with their structural stiffening with increasing pressure.

Here, we propose an iontronic flexible pressure sensor using a graded intrafillable architecture (GIA) consisting of unstable protruding microstructures of various heights, denoted as protrusions, that can easily buckle or bend upon compression, as well as complementary undercuts and grooves that allow for additional compressibility by accommodating the compressed protrusions. Such intrafillability plays a crucial role in promoting structural compressibility while simultaneously forming full contact area with the electrode, leading to a high sensitivity (220–3300 kPa$^{-1}$) and a high-pressure resolution (on the order of 10 Pa) over a broad pressure range up to 360 kPa. The sensors also show exceptional mechanical stability without noticeable fatigue over 5000 compression/release cycles at a high pressure of 300 kPa, or over 2000 bending/unbending cycles at a bending radius (R) of 6.5 mm. Given the large specific capacitance of the iontronic interface, the sensors exhibit a high-signal intensity and low noise when scaled down to the microscale (exemplified by 50 × 50 μm devices), such that they are expected to be an ideal candidate for a high-density sensing pixels array, which has also been experimentally demonstrated in this work. The sensors studied here will be highly useful when applied to robot

manipulation, pressure measurement in high-speed fluids (including aviation tests), where high-pressure resolution at high-pressure regimes is required. Additionally, the method of using GIA for highly sensitive pressure sensing over a broad pressure range, as a general mechanical design, will be effective when applied to other tactile sensors employing different material systems or different sensing mechanisms.

## Results

**Principles of GIA design and the sensing mechanism.** The term intrafillability we called here refers to the capability of a structure to accommodate its deformed portion by means of self-filling. Distinct from even surfaces, key morphological features of an intrafillable structure are the undercuts and grooves on the surface that can offer spaces to accommodate surrounding structures undergoing deformation. To elucidate the underlying mechanism by which a GIA produces remarkable sensitivity over a broad pressure regime, we investigated four representative microstructures by performing finite element analysis (FEA) (Fig. 1a): a hemisphere; a tilted pillar; an intrafillable pillar without gradient; and the GIA. In general, soft materials used for fabricating pressure sensors are incompressible, that is, the material preserves its volume during mechanical deformation. In the absence of interior voids (i.e., foam structure) or surface grooves, bulk structures made of incompressible materials exhibit high resistance to external pressure, yielding a low structural compressibility, which has been clearly exemplified by the hemisphere in Fig. 1a.

The tilted pillar in Fig. 1a represents a class of unstable protruding structures which can buckle down when compressed. However, the buckled pillar still presents rapidly increasing pressure-resistance once intimate contact is formed, performing like stable structures. Distinctly, the GIA, which has dense surface undercuts and grooves that can accommodate buckled protrusions, will improve the structural compressibility because of the following two aspects. First, the pillars buckle and start to fill into the surface undercuts upon compression, allowing for large deformations before they fully contact the bottom (see Fig. 1a, in the case of 50 kPa). Second, due to the uneven nature of the surface undercuts, there still exist some gaps between the buckled pillars and the surface undercuts after contact forms (see Fig. 1a, at 100 kPa). These gaps will be gradually filled as pressure grows, which allows the structure to be further compressed over high pressures (see Fig. 1a, from 50 to 400 kPa).

In addition to the surface undercuts, by further introducing a height gradient among the distributed protrusions, the electrode will come into shorter protrusions after the taller ones buckle, resulting in a gradually increasing contact area between the GIA film and the electrode over a wide pressure regime. Meanwhile, the initial contact area $A_0$ prior to applying pressure is accordingly minimized such that the normalized change of contact area $\Delta A/A_0$ of the GIA shows a substantial growth with increasing pressure in comparison to the other microstructures investigated (Fig. 1b). Such a significant increase of the normalized contact area subsequently escalates the specific capacitance due to the formation of EDLs at the iontronic GIA film/electrode interface, i.e., $C_{EDL}$, which is about 5–6 orders of magnitude higher than its non-iontronic counterparts. As illustrated in Fig. 1c, there are a large number of low molar positive and negative ion pairs distributed in the iontronic GIA film. When the voltage is applied, electrons on the electrode and the counter ions in the GIA aggregate within the contact area at a nanometer distance, elevating the capacitance[33,34].

A specific sandpaper with bulging grains and underlying holes (Supplementary Fig. 1a) was found to be an ideal template for

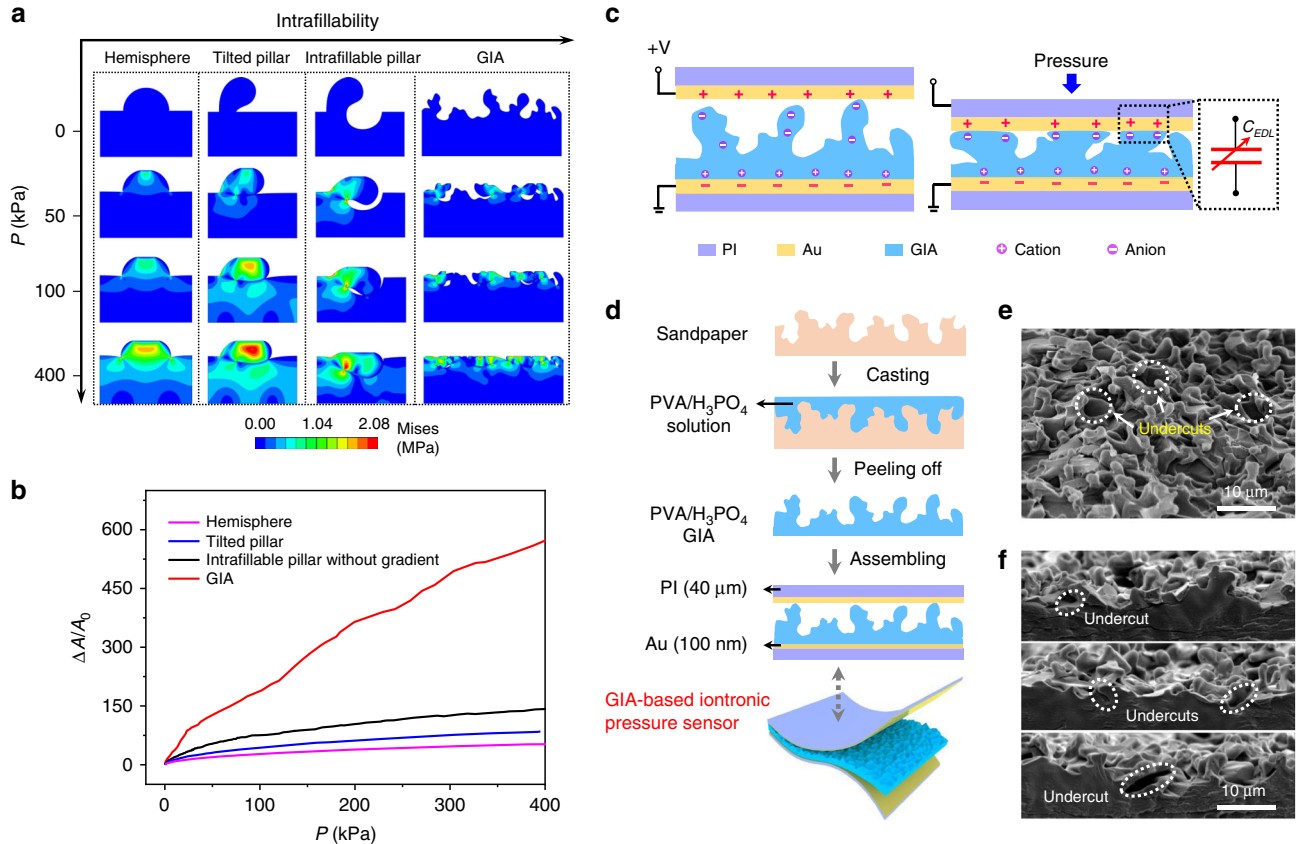

**Fig. 1 Principles of graded intrafillable architecture (GIA) design and the sensing mechanism. a** Stress distribution of simulation results for different architectures under pressures up to 400 kPa: a hemisphere; a tilted pillar, an intrafillable pillar without gradient, and the GIA. **b** Contact area variation between the dielectric layers with different architectures and an opposing electrode under a broad sensing range (0–400 kPa). **c** Schematic illustration for the functioning of the iontronic pressure sensor before and after applying pressure. **d** Schematic illustration of the preparation of a GIA-based iontronic pressure sensor. **e** A 45° tilt-view and **f** cross-sectional view SEM images of the PVA/$H_3PO_4$ GIA film, showing grooves and undercuts.

fabricating such a GIA film (Supplementary Fig. 1b). Figure 1d illustrates the fabrication process for the GIA-based iontronic pressure sensor, in which the GIA-based PVA/$H_3PO_4$ film (blue) and Au (brown) are employed as the dielectric and electrode, respectively. PVA/$H_3PO_4$ solution is casted on the sandpaper to be cured, followed by demolding. For the ionic PVA/$H_3PO_4$ film, the protrusions are molded from the holes of the sandpaper, while the grooves and undercuts are molded from the bulging grains. Figure 1e is a tilt-view scanning electron microscopy (SEM) image of the PVA/$H_3PO_4$ film demolded from the template, showing graded protrusions and surface grooves, as well as undercuts that are highlighted by white dashed circles. Cross-sectional view SEM images (Fig. 1f) also exhibit these protrusions and undercuts, confirming that the PVA/$H_3PO_4$ film is a legitimate GIA as we proposed. We also experimentally visualized the increase of contact area between PVA/$H_3PO_4$ GIA film and electrode under increasing pressure by dyeing the electrodes with ink (Supplementary Fig. 2a). Consistent with the FEA results, the contact area shows a substantial increase as the pressure increases, even in the high-pressure regime (Supplementary Fig. 2b).

**Sensing properties of the GIA-based iontronic pressure sensor.**
The sensitivity of a capacitive-type pressure sensor is defined as $S = \delta(\Delta C/C_0)/\delta P$, where $C$ and $C_0$ are the measured capacitance and the initial capacitance before applying pressure ($P$), respectively. A high sensitivity is achieved when a small change in pressure leads to a large capacitance change $\Delta C$. For non-

iontronic CPS, $\Delta C$ stems from the reduction in dielectric thickness and effective dielectric constant, which is usually limited to a few times that of $C_0$. Distinctly, $\Delta C/C_0$ of iontronic sensors can reach up to $10^6$ due to the formation of EDLs at the dielectric/electrode interface. The sensitivity of the GIA-based iontronic sensor studied here exhibits an unprecedentedly high value over a wide pressure range (Fig. 2a). The averaged sensitivity is $S_1 \sim$ 3302.9 kPa$^{-1}$ (or 3.3 Pa$^{-1}$) when the pressure is below 10 kPa, and is $S_2 \sim$ 671.7 kPa$^{-1}$ within the pressure range of 10–100 kPa. In the high-pressure regime (>100 kPa), the sensor exhibits a nearly linear response with a sensitivity $S_3 \sim$ 229.9 kPa$^{-1}$ up to 360 kPa. Surprisingly, the minimum sensitivity of our sensor is higher than the maximum sensitivity of any previously reported CPS[3,16,23–25,29,36]. Note that such a remarkable sensing performance is quite stable over a few samples (Supplementary Fig. 3). Three phases of the sensitivity can be briefly elucidated as follows. Before pressure is applied, the contact area of the GIA/electrode interface is substantially small (Supplementary Fig. 4), thus the initial capacitance $C_0$ is only several pF due to minimal EDL formation, and almost independent of test frequency (Supplementary Fig. 5a). As the pressure gradually increases to 10 kPa, the electrode come into contact with the tip of more taller protrusions (See experiments in Supplementary Fig. 2b at 10 kPa and FEA in Supplementary Fig. 4), resulting in a larger frequency-dependent EDL capacitance, which is an intrinsic characteristic of iontronic sensors (Supplementary Fig. 5b). The transition from a low initial capacitance ($C_0 \sim$ pF) to iontronic supercapacitance ($C \sim$ nF) produces an ultrahigh sensitivity in the low-pressure

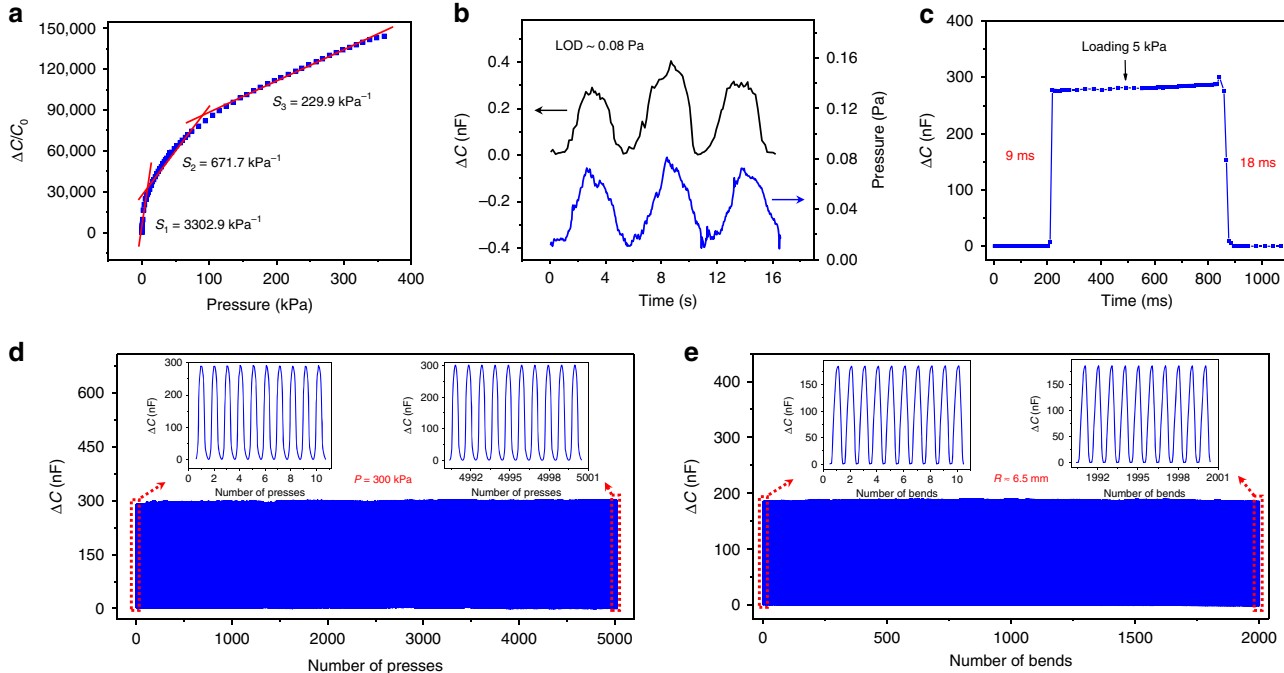

**Fig. 2 Sensing properties of the iontronic pressure sensor. a** Change of capacitance over the pressure range up to 360 kPa. **b** Limit of detection (LOD). **c** Response time at the frequency of 1 kHz. **d** Working stability tested over 5,000 cycles under a high pressure of 300 kPa. **e** Bending responses over 2,000 cycles at a bending radius (R) of ~6.5 mm.

regime. After that, when the pressure increases up to 100 kPa, buckling of taller protrusions takes place, followed by the intra-filling and contact with surface grooves (Fig. 1a, at 50 kPa). During this process, more microscale EDL capacitors are formed in parallel, leading to increasing capacitance. As the pressure further increases, the intrafilling advances by means of sub-stituting more interfacial gaps with buckled protrusions, and in the meanwhile the electrode comes to contact with protrusions of lower altitude, allowing for a steady escalation of EDL formation until most gaps are filled.

In addition, our sensor exhibits a low limit of detection (LOD) of 0.08 Pa as evidenced in Fig. 2b. To evaluate the dynamic response speed of the sensor, a weight of 10 g (equivalent pressure ~5 kPa) was gently placed on the pressure sensor followed by a quick release revealing a 9 ms response time and an 18 ms relaxation time (Fig. 2c), which are much faster than those of human skin (30–50 ms) and existing microstructured piezo-CPSs[16,17,22,37,38]. For flexible pressure sensors, high-mechanical durability under long-time or cyclic use also plays a crucial role in the reliable input-output relation. Repeated compression/release test over 5000 cycles with a peak pressure of 300 kPa was performed, and the sensor exhibits no signal drift or fluctuation (Fig. 2d) and negligible hysteresis (Supplementary Fig. 6) during the cyclic tests. In addition to the cyclic compression testing, we also investigated the flexibility of the device by testing signal stability under cyclic bends. Figure 2e suggests that our device maintains a remarkable mechanical robustness without noticeable fatigue after 2000 bending/unbending cycles with a bending radius of 6.5 mm. The high reliability under cyclic bending indicates that our sensor is a promising candidate for detecting bending-related body motion. We prepared a 3 mm × 15 mm sensor and evaluated its capability of detecting the bending of human finger and elbow joints (Supplementary Fig. 7a, b), confirming that different bending conditions can be easily distinguished according to the signal change. In addition, the sensor was found to be capable of detecting the pulse in a human

radial artery, and the signal shows a perfect waveform with quite strong characteristic peaks (Supplementary Fig. 7c). The afore-mentioned merits, including ultrahigh sensitivity, broad working range of pressure, short response/relaxation time, and high stability to mechanical loadings, all indicate the great potential of our device for various applications from health monitoring to wearable sensors, and for intelligent robotics.

Stability under different levels of relative humidity and different temperatures is also critical to real applications since humidity and temperature often affect the response of iontronic capacitive devices. Here, the iontronic pressure sensor is well packaged, such that the humidity change does not affect the signal (Supplementary Fig. 8a), while increasing temperature results in higher capacitance signal intensity (Supplementary Fig. 8b) due to the improved mobility of ions under higher temperature. The iontronic pressure sensor would, therefore, need to be calibrated to eliminate temperature effects when it is used beyond room temperature.

**Extremely high-pressure resolution**. A key advantage of our GIA-based iontronic pressure sensor is its sufficiently high sen-sitivity over a broad pressure regime, and its high-sensitivity under high pressure allows for a high-pressure resolution. Existing CPS or e-skins with surface microstructured dielectric often present low sensitivity or a saturated response above 50 kPa, and thus their pressure resolution is expected to be quite low. An ideal flexible pressure sensor, however, should detect tiny changes in pressure not only under low pressures but also under extremely high pressures, as schematically illustrated in Fig. 3a. The sensing performance of our device under three different reference pres-sures of $P_0 = 3$, 30, and 300 kPa is presented in Fig. 3b–d, respectively. For the test, the device was first compressed to a reference pressure, followed by consecutively adding three light-weight metal nuts, each weighing about 420 mg, which corre-sponds to an effective pressure increment of $\Delta P \sim 85$ Pa, and

capacitance was measured throughout the process. It shows that each pressure increment successfully leads to a stepped escalation of the capacitance with a swift response, and a steady signal is also confirmed for each pressure increment.

Several experiments were carried out to further demonstrate such extraordinary pressure resolution at high pressures. We first prepared a circle-shaped device with a radius of 8 mm and loaded it by placing it under a connecting rod (12.5 g, radius 8 mm) and three concrete bricks (~6.4 kg), i.e., equivalent to a reference pressure $P_0 = 320$ kPa, and we then gently put a pencil ($\Delta P \sim 300$ Pa), a layer of pencil shavings ($\Delta P \sim 40$ Pa), and feather-like fibers ($\Delta P \sim 18$ Pa) in sequence on top of the bricks (Fig. 3e). The

corresponding capacitance changes are displayed in Fig. 3f, which shows that each tiny pressure change can be precisely recorded and differentiated. In another experiment, a square-shaped sensor with a lateral size of ~1 cm was placed under the rear wheel of a car (2000 kg, which generates a pressure of a few hundred kilopascals), as indicated by the arrow in Fig. 3g. A bag of paper towels weighting only 1.7 kg was taken from the trunk of the car and then reloaded, and the capacitance changes were successfully detected as shown in Fig. 3h. Such a tiny pressure change cannot be discriminated by using iontronic sensors with other stable microstructures, such as microcones (Supplementary Fig. 9). This sensor is also capable of distinguishing the embarkation/

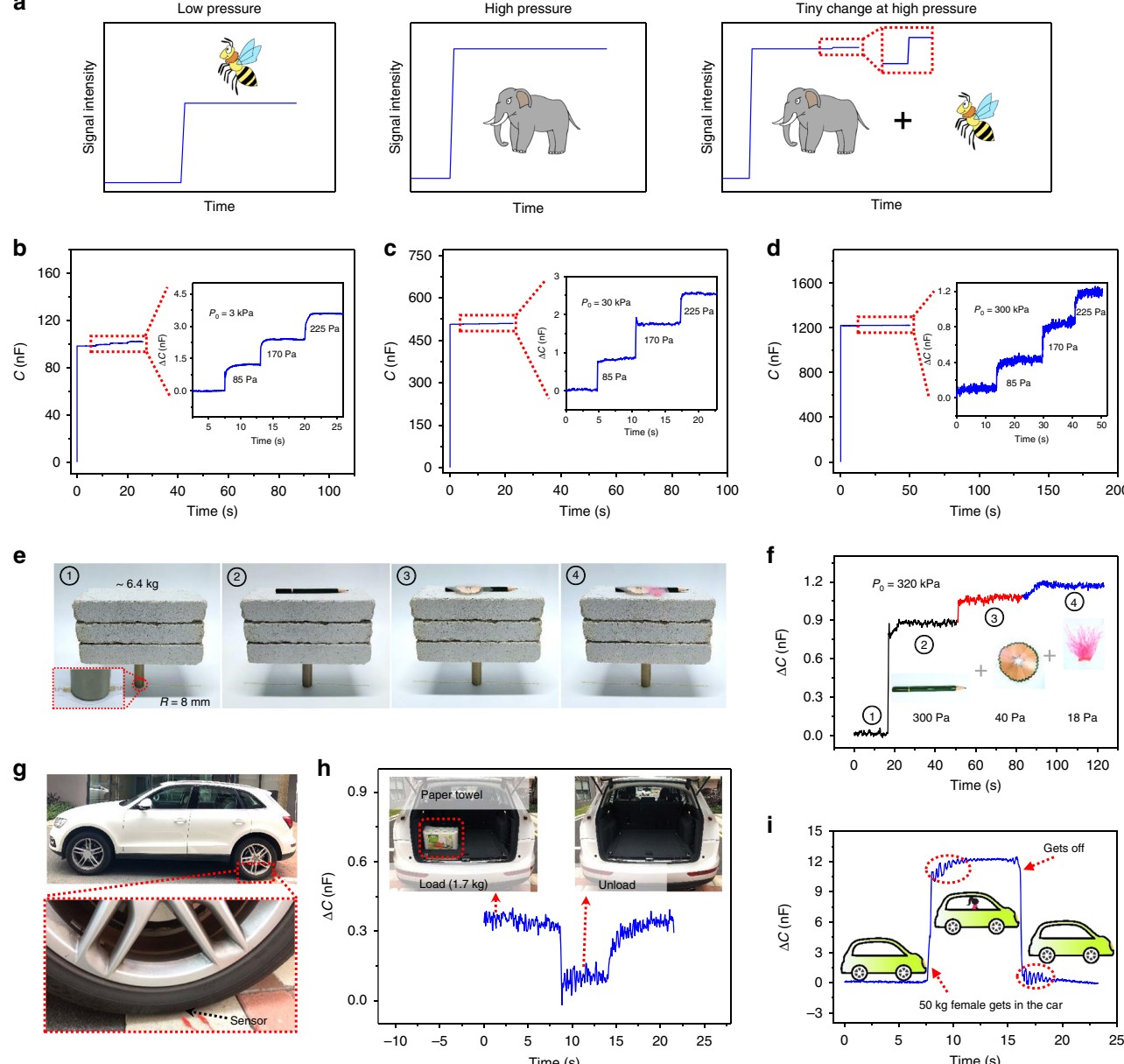

**Fig. 3 Extremely high-sensing resolution of the graded intrafillable architecture (GIA)-based iontronic pressure sensor. a** Schematic illustration of the response of the iontronic pressure sensor to low and high pressure, and detection of micro pressure under high pressure. **b–d** Detection of micro pressure under loading pressures of **b** 3 kPa, **c** 30 kPa, and **d** 300 kPa, respectively. **e** Detection of different micro pressure objects placed on three concrete bricks weighing 320 kPa. **f** Capacitance signals corresponding to panel **e**. **g** Experimental set-up of a car with a GIA-based iontronic pressure sensor bonded under a rear tire, the test frequency is 10 kHz. **h** Capacitance signals corresponding to a loaded, unloaded, and reloaded 1.7 kg bag of paper towels in the trunk of the car. **i** Capacitance signals corresponding to a 50 kg female passenger getting into and out of the car. The test frequency is 10 kHz.

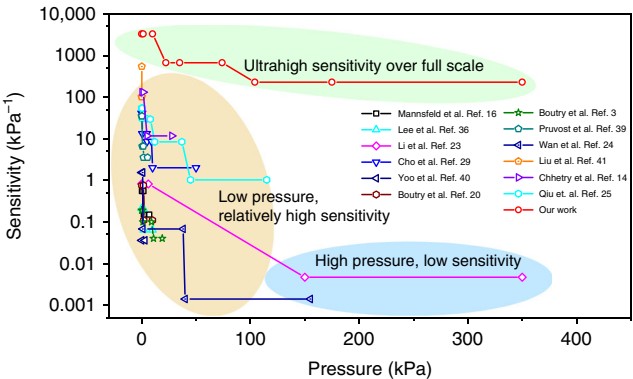

**Fig. 4** Comparison of the sensitivity of our pressure sensor with existing capacitive sensors.

debarkation of a 50 kg female passenger with a large capacitance change (~12 nF), as shown in Fig. 3i. Additionally, the measured signal also confirms that our sensor responses swiftly and sensitively to dynamic pressures (e.g., the circled oscillations in Fig. 3i reflect vibrations when the passenger gets into and out of the car), while providing stable output for static pressures with superior recoverability. Incorporating the morphological features of GIA, i.e., graded protrusions and surface undercuts, with ionic characteristics eventually leads to the prominent sensing behavior of our device over a sufficiently large pressure range.

As summarized in Fig. 4[3,14,16,20,23–25,29,36,39–41], our GIA-based iontronic pressure sensor shows an incomparably high sensitivity and an ultrabroad work range of pressure, outperforming existing piezo-CPS reported in the literature to the best of our knowledge. It is also worth noting that the sensitivity and working range of the GIA-based iontronic pressure sensor can be readily scaled by changing only the material moduli while maintaining other parameters the same (e.g., structures, ion density, and loading condition). According to Persson contact theory for randomly rough surface[42], the normalized contact area can be expressed as

$$\frac{\Delta A}{A_0} = \alpha \frac{P}{E} \quad (1)$$

where $\alpha$ is a geometric parameter, which depends only on surface morphology. Therefore, for a specific surface structure (i.e., $\alpha$ is constant), a larger modulus will render a higher sensing range while compromises the sensitivity simultaneously. This implication can also be supported by simulations of a specific GIA with different Young's moduli of $E_0$, $2E_0$ and $5E_0$ (Supplementary Fig. 10a). The corresponding normalized contact area $\Delta A/A_0$ as a function of applied pressure are shown in Supplementary Fig. 10b, which clearly suggests that the sensing range is increased when the material gets stiffer.

The high-pressure resolution of the sensor is of great importance. Although pressure-resolution is a critical parameter for conventional pressure sensors that are used in industry, it has been overlooked in e-skins or flexible pressure sensors. For reference, human skin can typically resolve a pressure difference of 7% under small pressures[43]. Our flexible sensor can recognize micro-pressure as low as 3 Pa under a low pressure of 3 kPa (Supplementary Fig. 11). More importantly, at an extremely high pressure of 320 kPa, our flexible sensor still presents a high-pressure resolution of 18 Pa, or 0.0056%, which is at least four orders of magnitude higher than that of human skin. Such a sensor will be quite useful in the accurate manipulation of heavy objects by robots, as well as pressure resolution for wind tunnel

tests. For example, the pressure mapping of a plane model in a high pressure wind tunnel test requires pressure sensors that can work over three atm with a minimal pressure resolution of 100 Pa. Although a few commercial sensors are already available, e.g., the PSI 8400 system[44], those sensors are often bulky, non-flexible, and need to be implanted in the plane body by drilling holes. Here, our sensors can resolve even smaller pressure change (18 Pa) under the high pressure required in wind tunnel tests, and are thus reasonably envisaged as flexible skins that are thin, cost-effective, and easily attached onto the curvilinear surfaces of a plane model for pressure mapping. Other applications where high flexibility and high-pressure resolution over a broad pressure range are required simultaneously may be found for our sensors.

**Spatial resolution of GIA-based micro-sensor arrays**. Direct-capacitive-type pressure sensors often suffer from low-specific capacitance and thus the signal is susceptible to noise when the devices are scaled down to the microscale. The iontronic sensor studied here is based on EDL capacitance or supercapacitance that allows for high-resolution pressure sensing with a high signal-to-noise ratio. Three micro-sensors with areas of 50 μm × 50 μm, 100 μm × 100 μm, and 200 μm × 200 μm were fabricated and their real-time responses to cyclic loading/unloading testing under a peak pressure $P = 10$ kPa are presented in Fig. 5a.

Owing to the high and stable capacitance density (Supplementary Table 1), our micro-sensors are able to maintain a reliable sensing performance with a high-signal intensity and negligible noise with an area of at least 50 μm × 50 μm. Therefore, our iontronic sensors can be used for electronics skins for high-resolution pressure mapping on the scale of ~100 μm. A micro-sensor array of 6 × 6 pixels over an area of 1.6 mm × 1.6 mm (Fig. 5b, pixel–pixel spacing is 150 μm) with each sensing pixel being a circular sensor of 60 μm in diameter was used to verify the pressure mapping on the sub-millimeter scale, and the micro-electrode is displayed in the inset of Fig. 5b. A tiny chip (0.2 g) was placed on the skin, partially covering it at two different locations and orientations, and a pressure of 50 kPa was applied for each. The location information was recorded and is precisely reflected in the corresponding pressure mapping images shown in Fig. 5c, d. It should be noted that no transistors were used in our micro-sensor array, while transistors are otherwise required for devices with high-spatial resolution, significantly simplifying the fabrication and reducing the cost.

The GIA-based iontronic sensor proposed in this work exhibits high sensitivity and high-pressure resolution over a wide pressure-sensing range due to the high intrafillability. Such a mechanical design is general for enhancing structural compressibility by intrafilling, and is also expected to be effective in other types of sensors, such as non-iontronic piezo-capacitive, piezo-resistive, and triboelectric sensors, for sensitive pressure sensing with a broad working range of pressure. For instance, we compared the sensitivity between non-iontronic GIA-based and hemispherical microstructure-based sensors (Supplementary Fig. 12a), and the GIA-based sensor exhibits a sensitivity significantly higher than that of the sensor with micro-hemispheres up to 360 kPa (Supplementary Fig. 12b). However, in the absence of EDL, the capacitance change of non-iontronic sensors are only limited to a few times the initial value. Therefore, the sensitivity for non-iontronic pressure sensors is much smaller than that of the iontronic pressure sensors (Supplementary Fig. 12c, d). It is also worth noting that ionic film (Young's modulus $E \sim 2.5$ MPa) is much softer than its non-ionic counterpart ($E \sim 67$ MPa) (Supplementary Fig. 13), which undoubtedly benefits the sensor and leads to higher sensitivity.

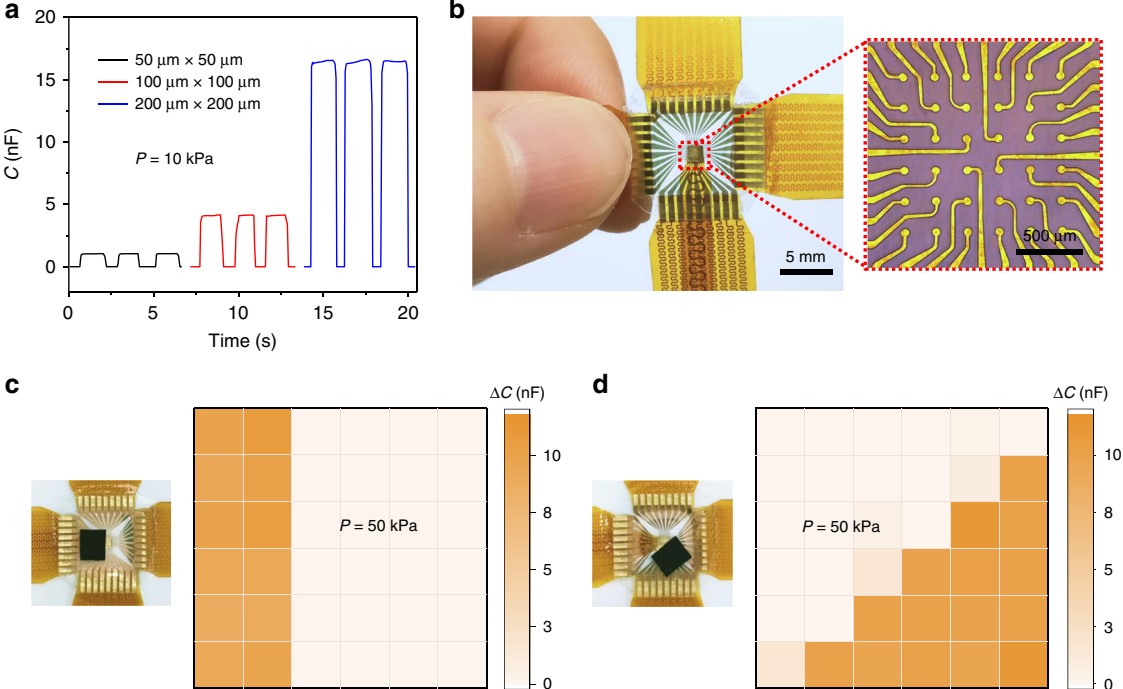

**Fig. 5 Spatial resolution of graded intrafillable architecture (GIA)-based micro-sensor arrays. a** Capacitance response of GIA-based micro-sensors with different sizes (square: 50 μm × 50 μm, 100 μm × 100 μm, and 200 μm × 200 μm) at the pressure of 10 kPa. **b** GIA-based micro sensor array (6 × 6 pixels) with circular sensing pixels. At right is the micro-electrode (pixel diameter, 60 μm; pixel spacing, 150 μm). **c**, **d** Pressure distribution mapping of the micro-sensor array in the shapes of **c** a rectangle and **d** a triangle (50 kPa).

## Discussion

Using a graded intrafillable architecture, in which easy-to-buckle protrusions and underlying undercuts and grooves that accommodate compressed microstructures are introduced, the ionic film effectively improves a sensor's structural compressibility via intrafilling. Capacitive-type sensors incorporating ionic film with such a GIA are able to simultaneously achieve a maximum sensitivity over 3 Pa$^{-1}$, and a large pressure-response range to at least 360 kPa, at which the sensitivity reaches over 200 kPa$^{-1}$. The sensors also exhibit a high-pressure resolution ~18 Pa (or 0.0056%) at a high pressure of 320 kPa. The high sensitivity over a broad pressure range stems from the micro-protrusions that buckles upon compression, as well as the intrafillable design that can accommodate deformed structures. The intrafillable structures, in the meanwhile, contribute to a rapid response speed of less than 10 ms, as well as high stability over 5000 loading/unloading cycles at a peak pressure of 300 kPa. We envisage that our GIA-based iontronic pressure sensor can find a verity of applications in health monitoring, electronic skins for tactile sensing, and pressure measurement in aerodynamics. We also believe that the intrafillable structure provides a general design strategy for other types of tactile sensors with a high sensitivity over a broad pressure range.

## Methods

**Finite element analysis**. FEA was performed using the commercial package ABAQUS 6.14. The PVA/H₃PO₄-based GIA was modeled as an incompressible neo-Hookean material with Young's modulus $E \sim 2.5$ MPa according to experimental measurement. The PI-Au electrode ($E \sim 3$ GPA) was simply treated as a rigid plate (not shown in Fig. 1a) and compressed downward. All contact interactions were assumed to be frictionless without penetration. Materials moduli were characterized in Supplementary Fig. 13. The complete simulation process is provided in the Supplementary Movie. The initial GIA/electrode contact area ($A_0$) in FEA was evaluated at a pressure of 100 Pa.

**Preparation of PVA/H₃PO₄ ionic films as the dielectric**. To realize the proposed GIA, we adopted a commercial sandpaper (roughness of no. 10000 #) as the template. First, 2 g of polyvinyl alcohol (PVA, Mw ~ 145,000, from Aladdin Industrial Corporation) was dissolved into 18 g of deionized water, followed by stirring at 90 °C for 2 h until it dissolved completely. After the PVA solution cooled to room temperature (22 °C), 1.65 mL H₃PO₄ (AR, ≥ 85%, Shanghai Macklin Biochemical Co., Ltd.) was added and stirred for 2 h. The PVA/H₃PO₄ solution was then poured onto the sandpaper and cured at room temperature for 12 h.

**Preparation of electrodes and the iontronic pressure sensor**. E-beam evaporator (HHV TF500) was used to deposit gold (100 nm) on the surface of flexible polyimide film (PI, thickness: 40 μm) to obtain the flexible PI-Au electrodes. The PI-Au film was then cut into certain sizes using a laser cutter (WE-6040), such as sizes of 7 mm × 7 mm, 3 mm × 3 mm, 3 mm × 15 mm, as well as circular shapes of 3 mm and 8 mm in diameters for different tests. The PVA/H₃PO₄ film was sandwiched between two PI-Au electrodes. Finally, the flexible iontronic pressure sensor was edge-packaged using 3M Scotch tapes.

**Fabrication of the micro-electrodes and micro-sensor arrays**. The fabrication process for the micro-electrodes is shown in Supplementary Fig. 14. First, a positive photoresist (Ruihong RZJ304) with a thickness of 3 μm on PI film was obtained by spin-coating at 2500 rpm for 30 s. After drying at 100 °C for 180 s, the PI film coated with the positive photoresist was photo-patterned by exposure to ultraviolet light for 8 s with the crosslinked region defined by the photomask. The film was then developed in developing solution (Ruihong, RZX3038) for 60 s, followed by hard baking at 120 °C for 90 s. Next, a Au film was evaporated on the patterned film by using E-beam evaporation. Finally, the photoresist in the unexposed areas was dissolved in a mixture of acetone and isopropanol with a volume ratio of 1:1. Utilizing the above mentioned method, we fabricated square micro electrodes with lateral sizes of 50 μm, 100 μm and 200 μm. The micro-sensor array (6 × 6 pixels, and each sensing pixel is a circle of 60 μm diameter, and the pixel–pixel spacing is 150 μm) was prepared by the same method except for using a different mask pattern. The PVA/H₃PO₄ film was then sandwiched between a PI-Au electrode and the micro-array electrode to prepare the micro-pixel array. The conductive thread electrodes were connected to the micro-sensor by using an anisotropic conductive adhesive.

**Characterization and measurements**. The microstructures of the PVA/$H_3PO_4$ film were characterized by using field-emission scanning electron microscope (FE-SEM, TESCAN). The capacitance was measured by using an LCR meter (E4980AL, KEYSIGHT). The external pressure was applied and measured accurately by using a force gauge with a computer-controlled stage (XLD-20E, Jingkong Mechanical Testing Co., Ltd). The stability tests were carried out using an iontronic pressure sensor with a size of 3 mm in diameter. The bending cycles of the flexible iontronic pressure sensor were evaluated using a smart stretching tester (WS150-100). The measurement of the radial artery pulse wave was carried out by attaching the iontronic pressure sensor to the wrist where the pulse could be detected at a testing frequency of 0.1 MHz. All other capacitance signals were tested at a frequency of 1 kHz unless otherwise specified, and all sensor size was set to 7 mm × 7 mm unless otherwise specified.

## Data availability
All relevant data sets generated during and/or analyzed during the current study are available from the corresponding author upon request.

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

## Acknowledgements
We thank the Materials Characterization and Preparation Center of SUSTech for assistance in the preparation of electrodes and support in materials characterization. This research was supported by the Innovation Committee of Shenzhen Municipality (Grant No. JCYJ20170817111714314), the National Natural Science Foundation of China (Nos. U1613204 and 51771089), the "Science Technology the Shenzhen Sci-Tech Fund (No. KYTDPT20181011104007), and the "Guangdong Innovative and Entrepreneurial Research Team Program" under Contract No. 2016ZT06G587.

## Author contributions
N.B., L.W. and C.G. designed the GIA structure. L.W. conducted the FEA. L.W. and C.G. wrote the paper. N.B. conducted the majority of the experiments. Q.W., J.D., Y.W., P.L., J.H. helped prepare the electrodes. G.L., Y.Z., J.Y., and K.X. participated in the discussion of experimental results. X.Z. and C.G. revised the manuscript. All authors reviewed and commented on the paper.

## Competing interests
The authors declare no competing interests.
