## [Peer Review File · Nature Communications]

Reviewers' comments:

Reviewer #1 (Remarks to the Author):

This manuscript reports a strategy for a high-performance capacitive pressure sensor using a special type of irregular surface architecture. The authors claimed the graded intrafillable architecture has provided an improved or even unprecedented sensing capability. However, this work lacks novelty in working mechanism and material modification, and the demonstration and analysis of wide pressure range sensing capability seem to be insufficient. Given that a variety of non-flat dielectric surface schemes already exist, and impressive sensing capability can be made generated by other approaches. Such a case, this work is not likely to provide the degree to which the results will stimulate new thinking in the field. I, therefore, feel that these findings would be better suited to publication in an alternative journal. I hope some of the comments may help the authors publish this work in the future.

1. The reason why the graded intrafillability architecture showed high sensitivity is explained by an increase in contact area when compressed. However, the reason for the high structural compressibility of this architecture isn't explained clearly. It is required to add more fundamental explaining to the results.
2. The advantage of this pressure sensor is that it has high sensitivity over a very wide range (Fig. 2a). Therefore, working stability tests (Fig. 2d) should be performed at higher pressure ranges.
3. In Fig. 2b, a limit of detection, LOD is 0.45 Pa. According to the author's explanation, GIA-type pressure sensors should have higher sensitivity under low-pressure ranges (Fig. 1b, 2a). Therefore, it is inconsistent with the experimental results. The author should describe why this discontinuity occurs.
4. The author should check for more errors in the entire manuscript.
5. Fig. 4 shows this pressure sensor's superiority of high sensitivity and ultrabroad work range of pressure in comparison with previous pressure sensors. However, there are several studies that also show better performance than references cited by the authors [Nature Communications, 6, 6269 (2015)] [ACS Appl. Mater. Interfaces, 11, 3438 (2019)] [Sensors, 18, 1001 (2018)]. It seems that the information about Fig. 4 was not researched sufficiently.
6. The main advantage of this pressure sensor is that it has high sensitivity over a very wide range. But the pressure sensor also shows extremely high sensitivity under low-pressure range (Fig. 2a). In order to increase the credibility of the experimental methods, the reasons for the pressure sensor's high sensitivity under low pressure should be explained more detailly.
7. Capacitive pressure sensors proposed by the author showed a sensitivity of less than 0.01kPa^{-1} at a 300kPa , while this pressure sensor show sensitivity of over 200kPa^{-1} under the same conditions (Fig. 4). The authors explained the result by a change in the contact area (Fig. 1b, Fig. S2), but the explanation is insufficient. In order to increase the credibility of the result, additional explanation through a computational analysis of the structure based on the properties of the material is required.

Reviewer #2 (Remarks to the Author):

The work by Bai et al. presents a capacitive pressure sensor (CPS) with high sensitivity over a broad pressure range. What has enabled such characteristics is the use of ionic elastomers in a combination with intra-fillable microstructures, which result in the contact area that increases monotonically with applied pressure over the wide pressure range. Given the fact that such architectures can easily be made from commodity items like sandpapers and that the proposed sensors exhibit high sensitivity over the broad pressure range, Reviewer would be in support of publishing this manuscript in Nature Communications eventually, but after minor revision addressing the following:

- 1) Add a brief discussion on why the proposed sensors exhibit three distinctive pressure ranges with different sensitivity values. Does this mean each operating pressure range is based on different working principles or their combinations?
- 2) Figure 1(a) Caption: "under an applied pressure of 400kPa" -> "under an applied pressure up to 400kPa"
- 3) Add a brief discussion on a batch-to-batch reproducibility of the proposed sensors. Do authors get similar sensitivity curves vs. applied pressure among the sensors made from different batches?

Reviewer #3 (Remarks to the Author):

In the manuscript entitled "Graded Intrafillable Architecture-based Ionic Pressure Sensor with Ultra-Broad-Range High Sensitivity", authors described an interesting method to establish an iontronic pressure sensing interface with a high device sensitivity at relative high pressure, which can potentially attribute to an interfacial GIA micro-structure of the ionic film. An ultra-broad sensing range is highly important for pressure sensing in many applications, and this manuscript is intended to address the challenges for its use in the flexible sensing area. However, there are still technical problems presented in its current format in this manuscript; in the view of the reviewer, those issues should be completely addressed before its acceptance to publication.

1. The key word in the title of ionic pressure sensor can be confusing. In fact, there are two types of pressure sensing mechanisms based on ionic materials, one is resistive-change based and the other relies on ionic/electronic electric double layer. Therefore, it is suggested to reflect the accuracy of the physical nature of this mechanism in the title by using iontronic or electric double layer-enabled pressure sensor.
2. The description of the sensing mechanism lacks of details. The analysis of the GIA structure shown in Figure 1a and 1b cannot exactly reflect the surface area change at different pressure. How to calculate A_0 in this simulation? A_0 should be zero if no pressure applied in theory, so the simulation should define the initial contact area, for instance, the contact area of different structures at the same tiny pressure, i.e. 10Pa. And what is the differences in structure between the intrafillable pillar and GIA? Only the difference in grain size? If so, the GIA structures with different grain sizes should be compared in the simulation.
3. In addition, is there a saturation point for this type of sensing? Can authors compare a planar iontronic interface to measure its unit-area capacitance using the same material system. It may indicate the sensitivity increase is caused by the surface roughness of GIA. More specific, how the properties of the GIA structure controls the sensitivity and range of the pressure sensor? For instance, how the grain size, the grain density, or the surface roughness affect the performance of the pressure sensor, e.g., device sensitivities? How controllable are those parameters? How does GIA compare to other surface structures such as micro-pyramid shapes, in addition to the hemispherical cap reported in this study?
4. A comparison of the sensitivity at high pressure should be done between different structures as a demonstration. For instance, can the normal structured pressure sensor sense the weight of the paper towel on a car?
5. A theoretic mechanical model would be helpful to explain why the GIA structure matters in contact area increase, in addition to the FEA efforts included.
6. Typically, 0-400kPa is not considered a very practically useful pressure range. The current work mainly focuses on a low pressure sensing range 0-200kPa (e.g., pulse measurement) or a high pressure sensing range 0-2MPa (e.g., dental or orthopedic applications). It would be nice this new

principle can be extended to a higher pressure range and illustrate its use for flexible sensing, given 400kPa range is not substantially higher than that of the existing devices.

7. Though mentioned in the introduction, several important earlier prior arts in the iontronic sensing area have been ignored, including:

B. Nie, et. al., "Iontronic Microdroplet Array for Flexible Ultrasensitive Tactile Sensing", Lab Chip, vol. 14, pp. 1107-1116, February 2014.

B. Nie, et. al, "Flexible Transparent Iontronic Film for Interfacial Capacitive Pressure Sensing," Advanced Materials, vol. 27(39), pp. 6055-6062, Oct 2015

8. Pg 14, Ln 244, authors claim that "For reference, human skin can typically resolve a pressure difference of 8% under small pressures", a reference is required to support this claim.

9. In Figure 2C, the response time of the device cannot be measured by finger pressing, as this action is very subjective and cannot reflect the true response time of the sensor.

Response to reviewers for the manuscript (NCOMMS-19-27329)

We would like to thank all reviewers for thoroughly reading our manuscript, and their constructive comments and concerns are carefully addressed and answered in a point-by-point manner. For convenience of the reviewers, the comments and suggestions are listed below in *blue font*, followed by our responses in normal black font. We also highlighted the corresponding modification in our revised manuscript in *red font*. We believe that the questions raised by the reviewers have highlighted areas in need of further attention and our modifications has contributed to the improvement of our manuscript, making it more suitable for the publication in *Nature Communications*.

Reviewer #1

This manuscript reports a strategy for a high-performance capacitive pressure sensor using a special type of irregular surface architecture. The authors claimed the graded intrafillable architecture has provided an improved or even unprecedented sensing capability. However, this work lacks novelty in working mechanism and material modification, and the demonstration and analysis of wide pressure range sensing capability seem to be insufficient. Given that a variety of non-flat dielectric surface schemes already exist, and impressive sensing capability can be made generated by other approaches. Such a case, this work is not likely to provide the degree to which the results will stimulate new thinking in the field. I, therefore, feel that these findings would be better suited to publication in an alternative journal. I hope some of the comments may help the authors publish this work in the future.

Response: The novelty in this work is focused on designing a graded intrafillable structure, which can significantly enhance the sensing performance of the device. In fact, the novelty on e-skins and flexible pressure sensors often comes from structure design. Although a variety of non-flat dielectric surfaces have already been reported, most of them consist of only “stable” structures such as micropyramid and microdome arrays, which suffer from saturated response at high pressures. By contrast, our sensor amalgamates both structure buckling and sunken surface grooves/undercuts, which enable full compressibility and high sensitivities over a broad pressure range by means of self-filling, and such a design has not yet been reported before.

1. The reason why the graded intrafillability architecture showed high sensitivity is explained by an increase in contact area when compressed. However, the reason for the high structural compressibility of this architecture isn't explained clearly. It is required to add more fundamental explaining to the results.

Response: To address the concerns of the reviewer, we have modified the manuscript by adding more detailed explanation to the sensing mechanism part.

Modification: Line 90-111, Page 5-6

“Distinct from even surfaces, key morphological features of an intrafillable structure are the undercuts and grooves on the surface that can offer spaces to accommodate surrounding structures undergoing deformation. To elucidate the underlying mechanism by which a GIA produces remarkable sensitivity over a broad pressure regime, we investigated four representative microstructures by performing finite element analysis (FEA) (Fig. 1a): (1) a hemispherical dome; (2) a titled pillar; (3) an intrafillable pillar without gradient; and (4) the GIA. In general, soft materials used for fabricating pressure sensors are incompressible, that is, the material preserves its volume during mechanical deformation. In the absence of interior void (*i.e.*, foam structure) or surface grooves, bulk structures made of incompressible materials exhibit high resistance to external pressure, yielding a low structural compressibility, which has been clearly exemplified by the hemispherical dome in Fig. 1a.

The titled pillar in Fig. 1a represents a class of unstable protruding structures which buckle down when compressed. However, the buckled pillar still presents rapidly increasing pressure-resistance once intimate contact is formed, performing like stable structures. Distinctly, the GIA, which has dense surface undercuts and grooves that can accommodate buckled protrusions, will improve the structural compressibility because of the following two aspects. First, the pillars buckle and start to fill into the surface undercuts upon compression, allowing for large deformations before they fully contact the bottom (see Fig. 1a, in the case of 50 kPa). Second, due to the uneven nature of the surface undercuts, there still exist some gaps between the buckled pillars and the surface undercuts after contact forms (see Fig. 1a, at 100 kPa). These gaps will be gradually filled as pressure grows, which allows the structure to be further compressed over high pressures (see Fig. 1a, from 50 to 400 kPa).”

2. The advantage of this pressure sensor is that it has high sensitivity over a very wide range (Fig. 2a). Therefore, working stability tests (Fig. 2d) should be performed at higher pressure ranges.

Response: A great point! Thanks for the suggestion, the working stability tested at a peak pressure of 300 kPa has been conducted, and the results show that there is no noticeable fatigue after 5,000 compression/release cycles, confirming that the pressure sensor has high mechanical stability. We have updated Fig. 2d in the revised manuscript accordingly. Such an improvement indicates that our sensor can be reliably used under high pressures.

Modification: Line 175-176, Page 9

“Repeated compressing/release test over 5,000 cycles with a peak pressure of 300 kPa was performed, and the sensor exhibits no signal drift or fluctuation (Fig. 2d)”

3. In Fig. 2b, a limit of detection, LOD is 0.45 Pa. According to the author's explanation, GIA-type pressure sensors should have higher sensitivity under low-pressure ranges (Fig. 1b, 2a). Therefore, it is inconsistent with the experimental results. The author should describe why this discontinuity occurs.

Response: We thank the reviewer for pointing the discrepancy out. We realized that this was caused by the resolution of force gauge of the Mechanical Testing Machine used before. After replacing a new force gauge with a much higher resolution, we found that the LOD of the pressure sensor is as low as 0.08 Pa. Few works have reported a LOD lower than 0.1 Pa. We have updated Fig. 2b in the revised manuscript accordingly.

Modification: Line 169, Page 9

“In addition, our sensor exhibits a low limit of detection (LOD) of 0.08 Pa as evidenced in Fig. 2b.”

4. The author should check for more errors in the entire manuscript.

Response: A native English speaker who is actually a professional language editor on scientific papers has went throughout our revised manuscript very carefully to minimize language errors.

5. Fig. 4 shows this pressure sensor's superiority of high sensitivity and ultrabroad work range of pressure in comparison with previous pressure sensors. However, there are several studies that also show better performance than references cited by the authors [Nature Communications, 6, 6269 (2015)] [ACS Appl. Mater. Interfaces, 11, 3438 (2019)] [Sensors, 18, 1001 (2018)]. It seems that the information about Fig. 4 was not researched sufficiently.

Response: We thank the reviewer for pointing out the missing references. To better serve the community, we have updated Fig. 4 in the revised manuscript by adding [ACS Appl. Mater. Interfaces, 11, 3438 (2019)] and [Sensors, 18, 1001 (2018)]. However, the pressure sensor reported in [Nature Communications, 6, 6269 (2015)] is resistive-type, which is different from capacitive pressure sensor and should not be reasonably compared here. Therefore, we do not include it in the revised manuscript because of different sensing mechanisms.

6. The main advantage of this pressure sensor is that it has high sensitivity over a very wide range. But the pressure sensor also shows extremely high sensitivity under low-pressure range (Fig. 2a). In order to increase the credibility of the experimental methods, the reasons

for the pressure sensor's high sensitivity under low pressure should be explained more detailly.

Response: Thanks for point this out. In fact, a higher sensitivity at a lower pressure is commonly seen in most existing e-skins and flexible tactile sensors, not specifically for our sensor. Such a phenomenon is determined by the structure. Although our structure has a broad pressure-responding range, it does not cause a change in this feature. Our experimental results (Supplementary Fig. 2) as well as simulation (Fig. 1b) have indicated that a GIA will result in a faster change in contact area under a lower pressure upon loading. Of course, it is possible to achieve a linear sensitivity over a wide pressure range by rationally designing novel microstructures, and this is confirmed in an on-going work of our team but out of the scope of this work. This point has been clearly explained in our response to the first point of Reviewer #2 (Line 154-168, Page 8-9 in the revised manuscript).

7. Capacitive pressure sensors proposed by the author showed a sensitivity of less than 0.01kPa^{-1} at a 300kPa , while this pressure sensor show sensitivity of over 200kPa^{-1} under the same conditions (Fig. 4). The authors explained the result by a change in the contact area (Fig. 1b, Fig. S2), but the explanation is insufficient. In order to increase the credibility of the result, additional explanation through a computational analysis of the structure based on the properties of the material is required.

Response: Direct computation of the specific capacitance of iontronic supercapacitor involves the diffusion of ions, interfacial conditions, as well as material properties, and is therefore extremely sophisticated. However, this problem can easily be simplified—the specific capacitance of iontronic supercapacitor has proven to be in direct proportion to the contact area due to the formation of EDL at the ionic gel-electrode interface. Therefore, the change in contact area can be adopted as an effective quantity for evaluating the capacitance change (and has actually be widely used). The reason for the high sensitivity of our GIA-based pressure sensor under high pressure is attributed to the high intrafillability that allows for more contact area even under large pressure. The FEA results presented in Fig. 1a-b clearly demonstrate that the GIA has a steady increase in contact area with the electrode as the pressure increases up to 400 kPa , while other microstructures do not behave similarly.

Reviewer #2:

The work by Bai et al. presents a capacitive pressure sensor (CPS) with high sensitivity over a broad pressure range. What has enabled such characteristics is the use of ionic elastomers in a combination with intra-fillable microstructures, which result in the contact

area that increases monotonically with applied pressure over the wide pressure range. Given the fact that such architectures can easily be made from commodity items like sandpapers and that the proposed sensors exhibit high sensitivity over the broad pressure range, Reviewer would be in support of publishing this manuscript in Nature Communications eventually, but after minor revision addressing the following:

Response: Thanks for the positive comments.

1) Add a brief discussion on why the proposed sensors exhibit three distinctive pressure ranges with different sensitivity values. Does this mean each operating pressure range is based on different working principles or their combinations?

Response: The three phases of pressure ranges are different. Under the low-pressure regime, taller protrusions do not buckle and EDL only forms at the interface between the protrusion tip and electrode. In the medium pressure regime, taller protrusions buckle down and EDL form between the surfaces of the buckled taller protrusions. In the high pressure regime, more EDLs form when the electrode contact with the shorter protrusion because of the altitude gradient.

Modification: Line 154-168, Page 8-9

“Three phases of the sensitivity can be briefly elucidated as follows. Before the pressure is applied, the contact area of the GIA/electrode interface is substantially small (Supplementary Fig. 4), thus the initial capacitance C_0 is only several pF due to minimal EDL formation, and is almost independent of test frequency (Supplementary Fig. 5a). As the pressure gradually increases to 10 kPa, the electrode come into contact with the tip of more taller protrusions (See experiments in Supplementary Fig. 2b at 10 kPa and FEA in Supplementary Fig. 4), resulting in a larger frequency-dependent EDL capacitance, which is an intrinsic characteristic of iontronic sensors (Supplementary Fig. 5b). The transition from a low initial capacitance ($C_0 \sim \text{pF}$) to iontronic supercapacitance ($C \sim \text{nF}$) produces an ultrahigh sensitivity in the low pressure regime. After that, when the pressure increases up to 100 kPa, buckling of taller protrusions takes place, followed by the intrafilling and contact with surface grooves (Fig. 1a, at 50 kPa). During this process, more microscale EDL capacitors are formed in parallel, leading to increasing capacitance. As the pressure further increases, the intrafilling advances by means of substituting more interfacial gaps with buckled protrusions, and in the meanwhile the electrode comes to contact with protrusions of lower altitude, allowing for a steady escalation of EDL formation until most gaps are filled.”

2) Figure 1(a) Caption: "under an applied pressure of 400 kPa" -> "under an applied pressure up to 400 kPa"

Response: Thanks for the suggestion, the expression of “under an applied pressure of 400kPa” has been changed to “under an applied pressure up to 400 kPa”.

3) Add a brief discussion on a batch-to-batch reproducibility of the proposed sensors. Do authors get similar sensitivity curves vs. applied pressure among the sensors made from different batches?

Response: Thanks for the suggestion. The reproducibility of the remarkable sensing performance has been confirmed by testing four samples. The results are presented in Supplementary Fig. 3.

Modification: Line 153, Page 8

“Note that such a remarkable sensing performance is quite stable over a few samples (Supplementary Fig. 3).”

Reviewer #3:

In the manuscript entitled “Graded Intrafillable Architecture-based Ionic Pressure Sensor with Ultra-Broad-Range High Sensitivity”, authors described an interesting method to establish an iontronic pressure sensing interface with a high device sensitivity at relative high pressure, which can potentially attribute to an interfacial GIA micro-structure of the ionic film. An ultra-broad sensing range is highly important for pressure sensing in many applications, and this manuscript is intended to address the challenges for its use in the flexible sensing area. However, there are still technical problems presented in its current format in this manuscript; in the view of the reviewer, those issues should be completely addressed before its acceptance to publication.

1. The key word in the title of ionic pressure sensor can be confusing. In fact, there are two types of pressure sensing mechanisms based on ionic materials, one is resistive-change based and the other relies on ionic/electronic electric double layer. Therefore, it is suggested to reflect the accuracy of the physical nature of this mechanism in the title by using iontronic or electric double layer-enabled pressure sensor.

Response: We thank the reviewer for correcting us. A more proper expression of “iontronic pressure sensor” has been adopted throughout the revised manuscript.

2. The description of the sensing mechanism lacks of details. The analysis of the GIA structure shown in Figure 1a and 1b cannot exactly reflect the surface area change at

different pressure. How to calculate A_0 in this simulation? A_0 should be zero if no pressure applied in theory, so the simulation should define the initial contact area, for instance, the contact area of different structures at the same tiny pressure, i.e. 10Pa. And what is the differences in structure between the intrafillable pillar and GIA? Only the difference in grain size? If so, the GIA structures with different grain sizes should be compared in the simulation.

Response: Initial GIA/electrode contact area (A_0) in all FEA is evaluated at pressure of 0.1 kPa, which has already been described in the section Materials and Method, *Finite element analysis*. The difference between the intrafillable pillar and GIA (graded intrafillable architecture) is that the later has a gradient distribution of the protrusions. To alleviate the reviewer's concern, we have changed the expression of "intrafillable pillar" to "intrafillable pillar without gradient" in the revised manuscript.

Modification: Line 95, Page 5

"we investigated four representative microstructures by performing finite element analysis (FEA) (Fig. 1a): (1) a hemispherical dome; (2) a titled pillar; (3) **an intrafillable pillar without gradient**; and (4) the GIA."

3. In addition, is there a saturation point for this type of sensing? Can authors compare a planar iontronic interface to measure its unit-area capacitance using the same material system. It may indicate the sensitivity increase is caused by the surface roughness of GIA. More specific, how the properties of the GIA structure controls the sensitivity and range of the pressure sensor. For instance, how the grain size, the grain density, or the surface roughness affect the performance of the pressure sensor, e.g., device sensitivities? How controllable are those parameters? How does GIA compare to other surface structures such as micro-pyramid shapes, in addition to the hemispherical cap reported in this study?

Response: The capacitance of the GIA iontronic sensor saturates when the pressure approaches to 1000 kPa. This saturation pressure may vary when the material modulus changes (see detailed discussion in Line 234-240 in revised manuscript). The sensitivity becomes much smaller beyond 360 kPa and thus it is not shown in Fig. 2a. For iontronic sensors, the key to enhance the sensitivity is to increase the gel/electrode contact area. Roughness plays a role in reducing the initial contact area A_0 . However, we cannot draw parallels between a rough surface and GIA. Theoretically, surface roughness is in direct proportional to length scale, while the deformation of the GIA is independent of length scale. For example, conventional microstructured surfaces (e.g. micropylramids and hemispheres) also have roughness (that can be made close to that of our GIA), but the sensor employing those microstructures exhibit much lower sensitivity. That is why our

simulation (also simulations in other papers) did not include any scale information but only focus on different structures. Similar conclusions can also be drawn in regard to grain size and density. Therefore, we did not consider surface roughness as a key parameter that affects sensing performance.

The performances of the sensors applying GIA, microcones (which are quite similar to micropylramids), and hemispheres are compared in Supplementary Fig. 10, indicating that the GIA performs better than others.

Modification: Line 234-240, Page 12

“It is also worth noting that the sensitivity and working range of the GIA-based iontronic pressure sensor can be readily scaled by changing only the material moduli while maintaining other parameters the same (e.g., structures, ion density, and loading condition). Based on dimensional analysis, the relative capacitance change $\Delta C/C_0$ (or contact area change $\Delta A/A_0$) is a non-dimensional function of normalized pressure P/E . Therefore, in order to achieve the same $\Delta C/C_0$, pressure P should scale with E . In other words, a smaller E yields a higher sensitivity but a relatively smaller sensing range, and vice versa.”

and Line 286-292, Page 15

“For instance, we compared the sensitivity between non-iontronic GIA-based and hemispherical microstructure-based sensors (Supplementary Fig. 10a), and the GIA-based sensor exhibits a sensitivity significantly higher than that of the sensor with micro-hemispheres up to 360 kPa (Supplementary Fig. 10b). However, in the absence of EDL, the capacitance change of non-iontronic sensors are only limited to a few times the initial value. Therefore, the sensitivity for non-iontronic pressure sensors is much smaller than that of the iontronic pressure sensors (Supplementary Fig. 10c and d).”

4. A comparison of the sensitivity at high pressure should be done between different structures as a demonstration. For instance, can the normal structured pressure sensor sense the weight of the paper towel on a car?

Response: Thanks for the suggestion, and we have added a demonstration with a sensor applying a microconed ionic gel layer. The sensor exhibited no response in sensing the weight of the paper towel on a car. This result agrees well with the sensitivity data shown in Supplementary Fig. 8.

Modification: Line 223, Page 12

“Such a tiny pressure change cannot be discriminated by using iontronic sensors with other

stable microstructures, such as microcones (Supplementary Fig. 8).”

5. A theoretic mechanical model would be helpful to explain why the GIA structure matters in contact area increase, in addition to the FEA efforts included.

Response: The current GIA has randomly distributed protrusions and surface undercuts that are molded from sandpaper. It is difficult to build a theoretical model for quantifying the contact area change as pressure grows. We are working on designing and programming GIA with regular patterns that may admit a theoretical analysis which may be reported in the near future.

6. Typically, 0-400kPa is not considered a very practically useful pressure range. The current work mainly focuses on a low pressure sensing range 0-200kPa (e.g., pulse measurement) or a high pressure sensing range 0-2MPa (e.g., dental or orthopedic applications). It would be nice this new principle can be extended to a higher-pressure range and illustrate its use for flexible sensing, given 400kPa range is not substantially higher than that of the existing devices.

Response: Thanks for the suggestion. The sensing range can be readily scaled by dimensional analysis which is added in the revised manuscript. We also want to point out the sensing range (0-400 kPa) studied in this work will be of great interest to aviation engineering, which highly needs sensors with a high pressure resolution up to 400 kPa.

Modification: Line 234-240, Page 12

“It is also worth noting that the sensitivity and working range of the GIA-based iontronic pressure sensor can be readily scaled by changing only the material moduli while maintaining other parameters the same (e.g., structures, ion density, and loading condition). Based on dimensional analysis, the relative capacitance change $\Delta C/C_0$ (or contact area change $\Delta A/A_0$) is a non-dimensional function of normalized pressure P/E . Therefore, in order to achieve the same $\Delta C/C_0$, pressure P should scale with E . In other words, a smaller E yields a higher sensitivity but a relatively smaller sensing range, and vice versa.”

7. Though mentioned in the introduction, several important earlier prior arts in the iontronic sensing area have been ignored, including:

B. Nie, et. al., “Iontronic Microdroplet Array for Flexible Ultrasensitive Tactile Sensing”, Lab Chip, vol. 14, pp. 1107-1116, February 2014.

B. Nie, et. al, “Flexible Transparent Iontronic Film for Interfacial Capacitive Pressure Sensing,” Advanced Materials, vol. 27(39), pp. 6055-6062, Oct 2015

Response: Done as suggested.

8. Pg 14, Ln 244, authors claim that “For reference, human skin can typically resolve a pressure difference of 8% under small pressures”, a reference is required to support this claim.

Response: The Weber fraction (the ratio between the smallest detectable difference and the reference value) of the force perception of human has been reported as around 7%. A reference has been cited to support this claim, and we have changed the value of “8%” to “7%” accordingly in the revised manuscript.

Ref: [Pang, Xiao-Dong, Hong Z. Tan, and Nathaniel I. Durlach. "Manual discrimination of force using active finger motion." *Perception & Psychophysics* 49.6 (1991): 531-540.]

9. In Figure 2C, the response time of the device cannot be measured by finger pressing, as this action is very subjective and cannot reflect the true response time of the sensor.

Response: Thanks for the suggestion. We retested the response time by placing/removing a weight of 10 g on the pressure sensor. The result has been updated in Fig. 2c in the revised manuscript.

Modification: Line 170-172, Page 9

“To evaluate the dynamic response speed of the sensor, a weight of 10 g (equivalent pressure ~ 5 kPa) was gently placed on the pressure sensor followed by a quick release revealing a 9 ms response time and an 18 ms relaxation time (Fig. 2c)”

Reviewers' comments:

Reviewer #1 (Remarks to the Author):

It seemed that the authors had addressed all points raised by reviewers, and the manuscript is in shape for publication.

Reviewer #2 (Remarks to the Author):

Reviewer believes authors properly addressed all the issues raised in the first round of review and is now ready for publication.

Reviewer #3 (Remarks to the Author):

In the revised manuscript entitled "Graded Intrafillable Architecture-based Iontronic Pressure Sensor with Ultra-Broad-Range High Sensitivity", the authors have attempted to address most of questions raised in my last review, including 1, 4, 6, 7 and 8. The revised manuscript could be further improved by providing detailed explanation on the theoretical analysis, as the current explanation still remains without solid evidence. On the other hand, the key innovation to provide high device sensitivity in a relatively high-pressure range proposed in this manuscript is of importance for flexible pressure sensing, and moreover, the results have shown the promise to solve this challenge. To achieve its broader impact, the authors could focus on the theoretical explanation on the specific structures and add more convincing evidence based on its current form. I would recommend a minor revision.

Response to Reviewer #3

Reviewer #3

In the revised manuscript entitled “Graded Intrafillable Architecture-based Iontronic Pressure Sensor with Ultra-Broad-Range High Sensitivity”, the authors have attempted to address most of questions raised in my last review, including 1, 4, 6, 7 and 8. The revised manuscript could be further improved by providing detailed explanation on the theoretical analysis, as the current explanation still remains without solid evidence. On the other hand, the key innovation to provide high device sensitivity in a relatively high-pressure range proposed in this manuscript is of importance for flexible pressure sensing, and moreover, the results have shown the promise to solve this challenge. To achieve its broader impact, the authors could focus on the theoretical explanation on the specific structures and add more convincing evidence based on its current form. I would recommend a minor revision.

Response:

The sensitivity and sensing range of GIA-based iontronic pressure sensor are intrinsically dependent on both intrafillable microstructures and material properties. A stiffer material will necessitates higher pressure to achieve full contact between gel and electrode, leading to a larger pressure sensing range, and vice versa. According to Persson contact theory on random rough surface [Persson BN. Contact mechanics for randomly rough surfaces. Surface Science Reports 61, 201-227 (2006)], the normalized contact area can be expressed as

$$\frac{\Delta A}{A_0} = \alpha \frac{P}{E}$$

where α is a geometric parameter which depends only on surface morphology. Therefore, for a specific surface structure, a larger E will produces a smaller contact area under the same pressure P , leading to a smaller capacitance change.

To provide more solid evidence of how material modulus affects the sensitivity and sensing range, we have performed two more simulations with different material moduli using the same GIA. The normalized contact area $\Delta A/A_0$ as a function of applied pressure is shown in Supplementary Fig. 9 which apparently suggests that larger modulus will render a higher sensing range while simultaneously reduces the sensitivity. Hence, such a GIA design can achieve even larger sensing range via increasing the material modulus. This design principle can be readily applied to other fields and find its applications in a relatively high-pressure range and reach a broader impact.

To alleviate the concern of the reviewer, we added the following sentences (marked in

red) in the revised manuscript

Line 237-246, Page 12:

“According to Persson contact theory for randomly rough surface⁴³, the normalized contact area can be expressed as

$$\frac{\Delta A}{A_0} = \alpha \frac{P}{E}$$

where α is a geometric parameter which depends only on surface morphology. Therefore, for a specific surface structure (*i.e.*, α is constant), a larger modulus will render a higher sensing range while compromises the sensitivity simultaneously. This implication can also be supported by simulations of a specific GIA with different Young’s moduli of $E_0, 2E_0$ and $5E_0$ (Supplementary Fig. 9a). The corresponding normalized contact area $\Delta A/A_0$ as a function of applied pressure are shown in Supplementary Fig. 9b, which clearly suggest that sensing range is increased when the material gets stiffer.”

REVIEWERS' COMMENTS:

Reviewer #3 (Remarks to the Author):

I would recommend the manuscript to be accepted as is now.

Response to Reviewer #3

Reviewer #3 (Remarks to the Author):

I would recommend the manuscript to be accepted as is now.

Response:

The authors appreciate the recommendation from the reviewer. Thank you for your time.